## METHOD

# ReadZS detects cell type-specific and developmentally regulated RNA processing programs in single-cell RNA-seq

Elisabeth Meyer[1,2†], Kaitlin Chaung[1,2†], Roozbeh Dehghannasiri[1,2] and Julia Salzman[1,2,3*]

†Elisabeth Meyer and Kaitlin Chaung contributed equally to this work.

*Correspondence:
julia.salzman@stanford.edu

[1] Department of Biochemistry, Stanford University, Stanford, CA 94305, USA
[2] Department of Biomedical Data Science, Stanford University, Stanford, CA 94305, USA
[3] Department of Statistics (by courtesy), Stanford University, Stanford, CA 94305, USA

## Abstract

RNA processing, including splicing and alternative polyadenylation, is crucial to gene function and regulation, but methods to detect RNA processing from single-cell RNA sequencing data are limited by reliance on pre-existing annotations, peak calling heuristics, and collapsing measurements by cell type. We introduce ReadZS, an annotation-free statistical approach to identify regulated RNA processing in single cells. ReadZS discovers cell type-specific RNA processing in human lung and conserved, developmentally regulated RNA processing in mammalian spermatogenesis—including global 3′ UTR shortening in human spermatogenesis. ReadZS also discovers global 3′ UTR lengthening in Arabidopsis development, highlighting the usefulness of this method in under-annotated transcriptomes.

**Keywords:** scRNA-seq, Differential RNA processing, Alternative polyadenylation, Untranslated regions

## Background

RNA processing is critical for understanding eukaryotic biology and disease. Differential RNA processing (RNAP) of the same gene, including kinetic rates of intron splicing, alternative polyadenylation (APA) sites, and 3′ untranslated region (3′ UTR) use, can regulate gene function and control RNA localization, stability, protein production, and translation efficiency [1–4]. Differential RNAP is widespread in eukaryotic genomes: genome-wide studies have shown that over 70% of mammalian protein-coding genes undergo APA [5] and intron retention [6]. Studies using bulk RNA sequencing have shown that RNAP is tissue-specific [7, 8] and is regulated during cell differentiation and proliferation [9–11]. On the other hand, altered RNAP has been increasingly linked to diseases from cancer to neurodegeneration and hematological disorders [12–14]. Identifying cell type-specific modes of RNAP regulation would lead to a deeper understanding of the mechanisms that determine the RNAP of particular genes, which would have major clinical implications.

Despite massive single-cell studies and a plethora of single-cell RNA sequencing (scRNA-seq) datasets generated over the past few years, scRNA-seq is currently underutilized in RNAP studies. Cell measurements are typically reduced to gene counts, limiting our understanding of the regulation of RNAP at the cell type and single-cell level. Technical limitations of scRNA-seq, such as low capture efficiency and high dropout rates, have led to the prevailing view that RNA is too sparsely sampled to measure alternative RNAP at single-cell level without imputation (as in scDaPars [15]) or pseudobulking (as in Sierra [16] and MAAPER [17]). Pseudobulking—aggregating reads from all cells within a cell type—increases power by amassing the sparse single-cell data together into bulk data. However, it also makes it impossible to measure heterogeneity within a pre-annotated cell type. Current computational methods employ various heuristics and lack statistical characterization, further limiting the possibility of targeted follow-up functional investigation of their discoveries. Moreover, current methods perform pairwise tests of differential RNAP, losing statistical power by requiring (n choose 2) tests for n cell types [15, 17] and limiting to analysis based on reads from the 3′ UTR alone.

The most common approach for detecting APA in bulk RNA-seq is peak calling [18], and this concept has been carried over to single-cell RNA-seq. "Peaks" are seen in poly(A)-primed RNA-seq data such as 10X due to preferred priming at a single 3′ end, which produces a distribution of insert lengths that is approximately normal after tagmentation [19] (Fig. S1). Peak calling-based methods—such as Sierra, MAAPER, and scDaPars (Table 1)—assign reads to one of several peaks, corresponding to 3′ UTR sites, and then measure the enrichment of peaks in different cell types or conditions. However, if two "peaks" originating from sites within one to two standard deviations of each other overlap, peak callers may not distinguish them. Further, biochemical error processes can cause failures of a strict parametric modeling of peaks.

Finally, most published algorithms for detecting APA rely on existing annotations, either a set of alternative transcripts or a list of polyadenylation sites previously documented for a given gene [20, 21]. While some poly(A) sites are annotated, a comprehensive annotation is still unavailable and very challenging from a computational perspective due to the difficulty of assigning reads from overlapping 3′UTRs, some of which may be lowly expressed [1–3, 22]. Even methods that rely solely on gene annotations are subject to similar bias due to incomplete annotation, particularly in organisms with poorly annotated genomes, but even in human [23]. The incompleteness of annotations limits the ability of annotation-reliant methods—such as Sierra, MAAPER, and scDaPars—to fully utilize single-cell resolved measurements and discover novel RNAP.

**Table 1** Comparison of existing APA methods and ReadZS

| Method | Sierra [16] | MAAPER [17] | scDaPars [15] | ReadZS |
|---|---|---|---|---|
| Single-cell resolved (no pseudobulking) | No | No | Yes | Yes |
| Imputation-free | Yes | Yes | No | Yes |
| Does not require gene annotations | No | No | No | Yes |
| Does not require peak calling | No | No | No | Yes |
| Can be used with continuous metadata, e.g., pseudotime | No | No | No | Yes |

To our knowledge, there is no annotation-free method to detect APA from either bulk or single-cell data.

## Results

### ReadZS enables statistical annotation-free detection of RNA processing in scRNA-seq

ReadZS is a computationally efficient, truly single-cell measure of RNAP. It does not use pseudobulking, imputation, or peak calling. As a true single-cell measure of differential RNAP, ReadZS can be integrated with continuous metadata such as pseudotime to identify developmentally regulated, continuous changes. It overcomes biases and the reduced statistical power inherent in annotation-dependent and peak calling approaches. It can detect differential RNAP at single-cell resolution that is regulated in any number of cell types and find regulated RNAP in developmental trajectories. ReadZS is applicable to 10X and other 3′ capture scRNA-seq methods as well as Smart-Seq2 (SS2).

We note that the ReadZS is predicted to detect primarily APA when applied to 10X data because reads in 10X data are enriched near the 3′ end of transcripts. 10X is a particularly ideal technology for profiling RNAP as it provides a much higher throughput compared to plate-based techniques (but at the cost of lower coverage for each cell), enabling profiling of RNAP across hundreds of cell types (even rare cell types). Moreover, 10X is designed to prime on poly(A) stretches of RNA, which are prevalent in introns and at the 3′ end of most cytoplasmic RNAs [24]. Reads arising from internal priming can still be used to detect regulated RNAP, and indeed ReadZS incorporates those reads as well. Indeed, ReadZS can be applied to data which is not 3′ enriched at all, such as SS2, and can still detect regulated RNAP without poly(A)-primed reads.

The major innovations of ReadZS include the following: (1) no reliance on exon, isoform, or gene annotation; (2) a purely statistical approach to analyzing read distributions that bypasses "peak calling" and all associated limitations (e.g., ad hoc minimum inter-peak distance, and only detecting cases with two peaks per gene [20]); (3) a truly single-cell-resolved score that can be integrated with other single-cell measurements such as developmental pseudotime; (4) a way to prioritize windows on the basis of effect sizes and quantifiable false discovery rate (FDR) for each set of calls; and (5) a very efficient, convenient, and reproducible workflow implementation based on Nextflow [25], with all needed packages and libraries pre-installed.

ReadZS first partitions each chromosome into genomic windows (treating each strand separately) and then summarizes the distributions of reads across each genomic window by giving a lower score to cells with reads closer to the downstream end of a window, and a higher score to cells with reads closer to the upstream end. This is achieved by reducing each uniquely mapped read to a rank within a genomic window across all cells ignoring metadata ("Methods," Fig. 1A). Ranks within each window are normalized to obtain read residuals using the population mean and standard deviation. The ReadZS value per cell per genomic window is defined by summing and scaling read residuals ("Methods"). Large negative (respectively, positive) ReadZS values mean that a cell's reads within a window are skewed upstream (resp. downstream) compared to the population average (Fig. 1A). In this paper, we analyze 5-kb windows in human and mouse and 1-kb windows in Arabidopsis. These lengths were chosen to capture variation in 3′ untranslated region (3′UTR) length, but this parameter is user-defined and flexible.

The ReadZS values for a given genomic window follow a normal distribution centered at zero under the null hypothesis that each cell has a statistically exchangeable [26] read distribution per window ("Methods"). Moreover, ReadZS is scaled such that if two or more subpopulations of cells exist within a sample, the expected value of the ReadZS will converge to a value that is a function of the cell population, independent of sequencing depth ("Methods").

The interpretable single-cell-resolved scalar value of the ReadZS means that its multivariate relationships with other covariates, such as pseudotime, can be evaluated without using cell type classification. Thus, ReadZS can detect regulated RNAP events that vary continuously with any measured covariate, such as space or time.

While the ReadZS method does not rely on annotation to detect windows with significantly regulated RNAP, after significant windows are called by ReadZS, their positions are intersected with annotation files to allow assignment of regulated RNAP events to a 3′ UTR, gene body, or unannotated region in order to enhance interpretability and downstream analysis.

When categorical cell type metadata is available, the cell type-level distributions of ReadZS values can be used to test whether median ReadZS scores per cell type and window are exchangeable (Fig. 1A). After multiple hypothesis testing correction, the pipeline calls genomic windows that are differentially processed across any number of cell types, with no need to pre-specify pairs of cell types to compare, a unique characteristic of ReadZS missing in all previous methods. This single test to detect differences among n cell types increases power compared to pairwise differential testing as $O(n^2)$ fewer tests are required. The range in median ReadZS by cell type defines an "effect size" which can be used to systematically prioritize genomic windows with larger variation in RNAP for subsequent analysis.

We now present a technical study of ReadZS based on real scRNA-seq data sets: (1) ReadZS rediscovers and extends known cell type-specific regulation of RNA processing

(See figure on next page.)

**Fig. 1** Overview of the ReadZS. **A** Read positions are ranked in equal-sized genomic bins, separated by read strand. Within each genomic window, the read distribution for each cell is summarized by a weighted, normalized function of read positions ("Methods"). With metadata, cell type-specific RNAP can be detected by finding windows with significantly different median ReadZS by cell type. Continuous metadata such as pseudotime enables discovery of multivariate relationships between ReadZS and metadata. GMM-based peak detection is used to compare read distributions with annotated 3′ UTRs. **B** Read distributions in the genomic window with largest effect size in HLCA P3 when requiring minimum 10 counts in 20 cells, which overlaps the genes *CORO1B* and *PTPRCAP*. Peaks in significant windows called by the GMM (see text) are starred. **C** *CALM1* is called in both P3 and P2 as having cell type-specific differences in RNAP. Peaks in significant windows called by the GMM are starred; peaks are called across all cell types. In *CALM1*, the peaks are 254 and 285 bp from the closest downstream 3′ UTR. **D** *KLF6* is called in both P3 and P2 as having cell type-specific differences in RNAP. The relative rank of each cell type (ranked by highest to lowest median ReadZS) is shown for each participant. Peaks in significant windows called by the GMM are starred; peaks are called across all cell types. **E** The ReadZS is technically reproducible across cell types in the 3 HLCA participants; *p*-values, computed via simulation (see "Methods"), show strong ReadZS concordance in all pairs. **F** Histogram and CDFs of the distribution of distances from GMM-called peaks to closest downstream annotated 3′ UTR in HLCA P3; lines denote the 25th, 50th, and 75th quantile, respectively. Distance distributions are compatible with expectation from 10x library construction. **G** Above: read distributions in the genomic window with largest effect size in HLCA P3 when requiring minimum 5 counts in 10 cells, which overlaps the genes *CATIP1*, respectively. Below: ReadZS distribution for the genomic window overlapping *CATIP* in four cell types from HLCA P3

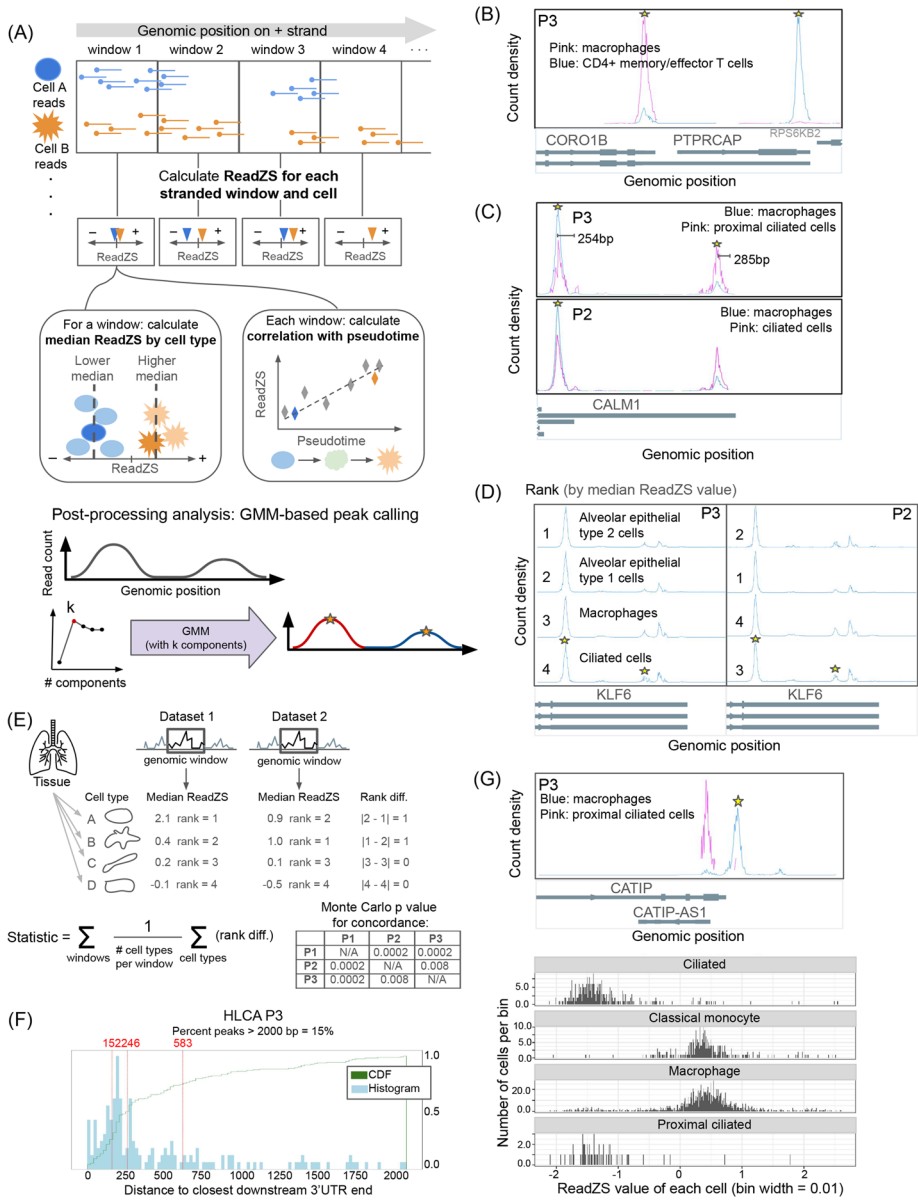

**Fig. 1** (See legend on previous page.)

in human lung; (2) ReadZS provides significantly concordant calls of RNAP across different biological replicates; (3) a post-facto peak calling on genomic windows with significant ReadZS variation between cell types shows the inferred poly(A) priming sites are enriched with known 3′ UTR ends, while discovering a significant number of sites that cannot be explained by priming from annotated 3′ UTRs, highlighting the need for annotation-free methods; (4) ReadZS rediscovers and extends known regulation of 3′ UTR length in human and mouse spermatogenesis; (5) ReadZS discovers developmentally regulated RNAP in Arabidopsis root, including global 3′ UTR lengthening; and (6) ReadZS results are consistent with the recently published algorithms Sierra [16] and MAAPER [17].

## ReadZS rediscovers and extends known regulation of RNA processing

We applied ReadZS to 10X data of non-tumor samples from three participants in the Human Lung Cell Atlas (HLCA), together encompassing 57 cell types in the lung and blood [27, 28]. We chose this dataset because it is deeply curated and thought to define all existing subtypes of cells in the lung. Consistent with [27], in this manuscript, we used participant 3 (P3)—the most deeply sequenced individual—as the primary participant and P1 and P2 individuals to validate our discoveries on P3. We ran the ReadZS pipeline on data from each participant separately. We required at least 10 counts in 20 cells in at least two cell types to calculate the ReadZS. ReadZS was calculable in 454 genomic windows (across 432 genes) in P3, from which 94 windows (20.7%, in 94 genes) were called as having significant cell type-specific RNAP (FDR < 0.05, "Methods," Additional file 2: Table S1). Similar proportions of significant windows were found in the two other participants (Additional file 1: Fig. S2).

To illustrate ReadZS discoveries in the lung, we examined the cell type-specific windows with the highest effect sizes (defined as the range of medians of ReadZS across cell types within an individual) in HLCA P3. The highest effect size reflects two 3′ UTRs in overlapping genes within a single 5-kb genomic window, a rare event in the human genome: *PTPRCAP*, a transmembrane phosphoprotein, and *CORO1B*, an actin-binding protein that controls cell motility (Fig. 1B). For this genomic window, we illustrate differential ReadZS values using the two cell types with sufficient reads to calculate median ReadZS values (≥10 counts in ≥20 cells). CD4+ memory/effector T cells dominantly express *PTPRCAP* whereas lung macrophages dominantly express *CORO1B*. This difference creates a dramatic shift in read distributions, demonstrating that ReadZS indeed detects genomic windows with large cell type-specific differences in read distribution.

Windows overlapping the genes *RPLP1*, *NEAT1*, and *SRSF7* were among the top 10 significant windows as ranked by effect size. In *RPLP1*, a component of the 60s subunit of the ribosome, intronic reads are significantly enriched in CD8+ memory/effector T cells relative to proliferating basal cells (Additional file 1: Fig. S3). *NEAT1* is a long noncoding RNA involved in nuclear paraspeckle assembly and undergoes extensive splicing and APA, but its isoforms have unknown functions [29]. Differential RNAP of *NEAT1* has important biological consequences, as higher expression of a longer isoform of *NEAT1* has been associated with poor prognosis in breast cancer, though the mechanism remains unknown [30]. For each significant window called by ReadZS, we performed peak calling to identify potential (known or unannotated) polyadenylation sites, by fitting a Gaussian mixture model (GMM) to the distribution of the reads from the entire dataset across that window ("Methods"). One peak detected by GMM postprocessing in *NEAT1* coincides with an annotated end and one not annotated—this could reflect either unannotated APA or internal priming resulting from alternative splicing (Additional file 1: Fig. S3). A similar phenomenon is observed in *SRSF7*, a master splicing regulator implicated in tumor progression [31]. For this gene, CD4+ T cells exhibit an unannotated GMM-called peak that could be evidence of unannotated alternative splice variants or unannotated APA (Additional file 1: Fig. S3).

In *CALM*1, the window with the next-highest effect size after the four windows listed above, two mixture components called by the GMM each correspond to an annotated 3′

UTR, which are differentially represented in proximal ciliated epithelial cells and macrophages (Fig. 1C). *CALM1* regulates calcium signaling and is known to undergo APA; in mouse, its long isoform is primarily expressed in neural tissue, and its 3′ UTR has been shown to control localization and be functionally essential [32]. ReadZS extends this finding and reveals significant cell type-specific regulation of 3′ UTR length of *CALM1*, specifically that proximal ciliated cells have highest use of the longest 3′ UTR, consistent with the idea that the long isoform of *CALM1* is related to excitatory cell function [32].

Differential RNAP in *KLF6*, a tumor suppressor regulating transcription [33] involves alternative regulation of the 3′ UTR of *KLF6* in ciliated cells and macrophages compared to other cell types such as alveolar fibroblasts (Fig. 1D). According to the gene annotation of *KLF6*, these reads support the use of unannotated 3′ UTRs which are predicted to change the protein coding potential of *KLF6*, albeit at lower frequency than the dominant priming site. Because these variants modify the 3′ UTR, they have unknown impacts on translation and thus protein abundance. Together, these examples illustrate the unique power of ReadZS to identify regulation including at unannotated 3′ UTR sites.

### ReadZS calls are consistent across biological replicates

To assess the ability of bioinformatics to distinguish multifactorial biochemical errors introduced during library preparations, benchmarking analysis based on real data is preferred to simulated data [34]. As the HLCA dataset encompassed three individuals, we first measured the concordance of the ReadZS calls by assessing the overlap of results between HLCA individual pairs. Because different sampling depths across cell types could impact concordance analysis for called windows, we restricted to windows with calculable ReadZS in both participants for each pairwise comparison. Restricting to the 245 windows that have calculable ReadZS in P3 and P2, 29 were significant in both individuals (hypergeometric $p$-value $< 5E-08$). Similarly, the P3-P1 and P2-P1 comparisons showed significant overlaps ($p$-value $< 0.002$).

We further assessed the concordance in the directionality of ReadZS by comparing the ordering of median ReadZS values for the same genomic window in different data sets. For example, if the median ReadZS value of a genomic window is higher for a certain cell type than for other cell types in one dataset, we expect this relative ordering to be consistent in other datasets (Fig. 1E). We calculated the consistency of median ReadZS order using a multivariate metric based on the Spearman footrule ("Methods"). The concordance of directionality of the ReadZS value per cell type was highly significant compared to random ordering of cells ($p$-value $< 0.005$ for P1-P2 and P2-P3; $p$-value $< 0.01$ for P1-P3).

### Statistically identified read peaks in windows with cell type-specific RNAP are enriched for known poly(A) sites and predict new APA sites

We further evaluated the biological relevance of ReadZS calls by calculating the fraction of windows called by ReadZS as having cell type-specific RNAP that can be explained by poly(A) priming at annotated 3′ UTRs. Given the 3′ bias of 10X sequencing, the detected differential RNAPs in 10X data are expected to be enriched at annotated 3′ UTRs.

Indeed, 81 of 94 (86.2%) ReadZS-significant windows in P3 overlap with at least one 3′ UTR annotation (Additional file 2: Table S1).

To further assess the biological properties of cell type-specific windows, we performed a statistical postprocessing step with Gaussian Mixture modeling (GMM) to define regions of high read density in the cell type-specific windows (Fig. 1A). The GMM summarizes elevated read density within a window by modeling it as a mixture of Gaussians and the means of the components can be defined as the peak locations ("Methods").

We quantified the distance between the means of the fitted GMM and the nearest annotated downstream 3′ UTR end, conditioning on this distance being less than 2kb because of known intronic priming. In HLCA P3 data, the median distance from the GMM means to the nearest annotated 3′ UTR is 286 bp, consistent with the average insert length of ~350bp in 10X libraries (Fig. 1F; Additional file 1: Fig. S4) [35]. This supports the idea that ReadZS-significant windows and the GMM approach to identify peaks primarily recover annotated 3′ UTRs even though no annotation was used in picking the significant windows. One of the novel 3′ UTRs was identified in *CATIP* in lung macrophages (Fig. 1G). The genomic window intersecting this gene was called as significant in HLCA P3 data when we reduced the minimum required number of counts per cell and cells per cell type to 5 and 10, respectively. Indeed, the window overlapping *CATIP* has the largest ReadZS effect size out of all genomic windows, suggesting there is strong cell type-specific regulation of this 3′UTR. CATIP plays a role in actin polymerization and organization of cilia, but the role of its different isoforms is not known [36].

We note that for genomic windows containing two peaks within an insert-size-distance of each other, the ReadZS cannot distinguish between variation in 3′UTR length versus variation in the length of the poly(A) tail. In other words, it is possible that a downstream peak could be caused by priming at the end of a longer poly(A) tail, while the 3′UTR length remains the same. However, if peaks are separated by more than the insert length (~350bp), the different peaks cannot be explained by differential poly(A) tails as the insert is smaller than the interpeak distance.

### Single-cell resolution of ReadZS reveals evolutionarily conserved, developmental post-transcriptional regulation in mammals

Global 3′ UTR shortening during mouse spermatogenesis is a well-documented but incompletely understood post-transcriptional regulatory program [37, 38]. We tested if ReadZS could detect global changes in 3′ UTR length from scRNA-seq data of mouse and human spermatogenesis [39, 40]. In this study, the authors used 10X sequencing to identify gene expression patterns in over 62,000 human and mouse spermatogenic cells, and thereby assigned each cell a pseudotime reflecting its stage of differentiation from stem cell to spermatid. For each genomic window, we calculated the correlation between estimated pseudotime and ReadZS value (Fig. 2A). We should note that this type of analysis is impossible to do with other methods that use pseudobulking (e.g., Sierra, MAAPER) or are limited to comparisons between clusters (e.g., scDaPars). In human, restricting to the 563 windows overlapping annotated 3′ UTRs ("Methods"), 93 windows had significant correlation to pseudotime (|Spearman's correlation| > 0.3, Bonferroni-corrected *p*-value < 0.05; Additional file 3:

Table S2). Fourteen out of 93 (15%) windows were positively correlated, consistent with 3′ UTR lengthening, and 79 (85%) were negative, consistent with global shortening (hypergeometric *p*-value < 1E−20 for enrichment of negatively correlated windows). In mouse, restricting to the 310 windows overlapping annotated 3′ UTRs, 3 (1%) significant windows had signs consistent with 3′ UTR lengthening and 307 (99%) had signs consistent with shortening (hypergeometric *p*-value < 0.017). This finding is consistent with work showing that 3′ UTRs globally shorten during mouse spermatogenesis [20, 28, 37]; we are not aware of studies that have reported this phenomenon, or the genes we identify as regulated during spermatogenesis, in humans.

To test if there is evolutionary conservation of mammalian genes undergoing regulated changes in 3′ UTR length, we matched and found 374 genomic windows (in 314 genes) annotated with the same gene in both mouse and human ("Methods"). Fifty-six of 374 (15%) of window pairs were significantly correlated with pseudotime in both human and mouse, significantly more overlap than expected by chance (hypergeometric *p*-value = 0.006). Forty-two out of 56 pairs had the same sign of correlation, corrected for gene direction (hypergeometric *p*-value = 1.27E−6). For example, ZFAND6, a zinc finger protein implicated in the pathophysiology of diabetes but not studied in spermatogenesis [41], shows high conservation across vertebrates in the 3′ UTR. Indeed both human and mouse exhibit similar patterns of 3′ UTR shortening in spermatogenesis (Fig. 2B). Mouse read distributions further support an unannotated 3′ UTR (indicated by the right-most dotted red line in Fig. 2B). The significant overlap in magnitude and direction of significant correlations between human and mouse supports the finding that ReadZS detects unreported, evolutionarily conserved regulation of RNAP during spermatogenesis.

*ARPP19*, a gene known to be a mitotic regulator but with unreported 3′ APA regulation [42, 43], has the largest negative correlation with pseudotime in human, reflecting 3′ UTR shortening. The second highest magnitude correlation is in *S100A10*, a gene studied in the immune system but with unknown function in sperm [44]. ReadZS detects a shift in RNAP over time, but peaks in the detected window are overlapping

(See figure on next page.)

**Fig. 2** ReadZS detects developmentally regulated RNAP in human and mouse spermatogenesis. **A** The ReadZS detects a global trend of 3′ UTR shortening in both human (left) and mouse (right) spermatogenesis datasets, indicated by significant negative correlation between ReadZS and pseudotime. Significance is defined as |Spearman's correlation| > 0.3 and Bonferroni-corrected *p*-value < 0.05. Histogram bin width = 0.02. **B** The ReadZS reveals evolutionarily conserved 3′ UTR regulation in human and mouse. Left: windows containing the 3′ end of *ZFAND6* were significantly correlated with pseudotime in both human (Spearman's correlation = −0.341, Bonferroni-corrected *p*-value < 1E−39) and mouse (Spearman's correlation = −0.757, Bonferroni-corrected *p*-value < 1E−84). Vertical red lines indicate peak positions. Right: the 3′ UTR region of *ZFAND6* in human shows high conservation with other vertebrates (UCSC Genome Browser). Red lines correspond to peak positions from the left plot. **C** The ReadZS discovers fine-scale developmental regulation of RNAP in human spermatogenesis, including in regions where neighboring APA sites create highly overlapping peaks. Note that the scatterplots show ReadZS before sign correction based on gene direction. Left, top to bottom: windows with significant correlation between ReadZS and pseudotime within the first 0 to 25% of pseudotime: OAZ1, which is both significantly correlated over all pseudotime (Spearman's correlation = 0.131, Bonferroni-corrected *p*-value = 1.5E−06) and within the first 0 to 25% of pseudotime (Spearman's correlation = 0.59, Bonferroni-corrected *p*-value < 1E−55), and *MED21*, which is only significantly correlated when restricting the first 0 to 25% of pseudotime (Spearman's correlation = −0.535, Bonferroni-corrected *p*-value < 1E−23), not across all of pseudotime. Right, top to bottom: the windows with the two highest correlation values when calculated over all of pseudotime: *ARPP19* (Spearman's correlation = 0.74, Bonferroni-corrected *p*-value < 1E−223) and *S100A10* (Spearman's correlation = 0.666, Bonferroni-corrected *p*-value < 1E−88). Red lines highlight peak positions

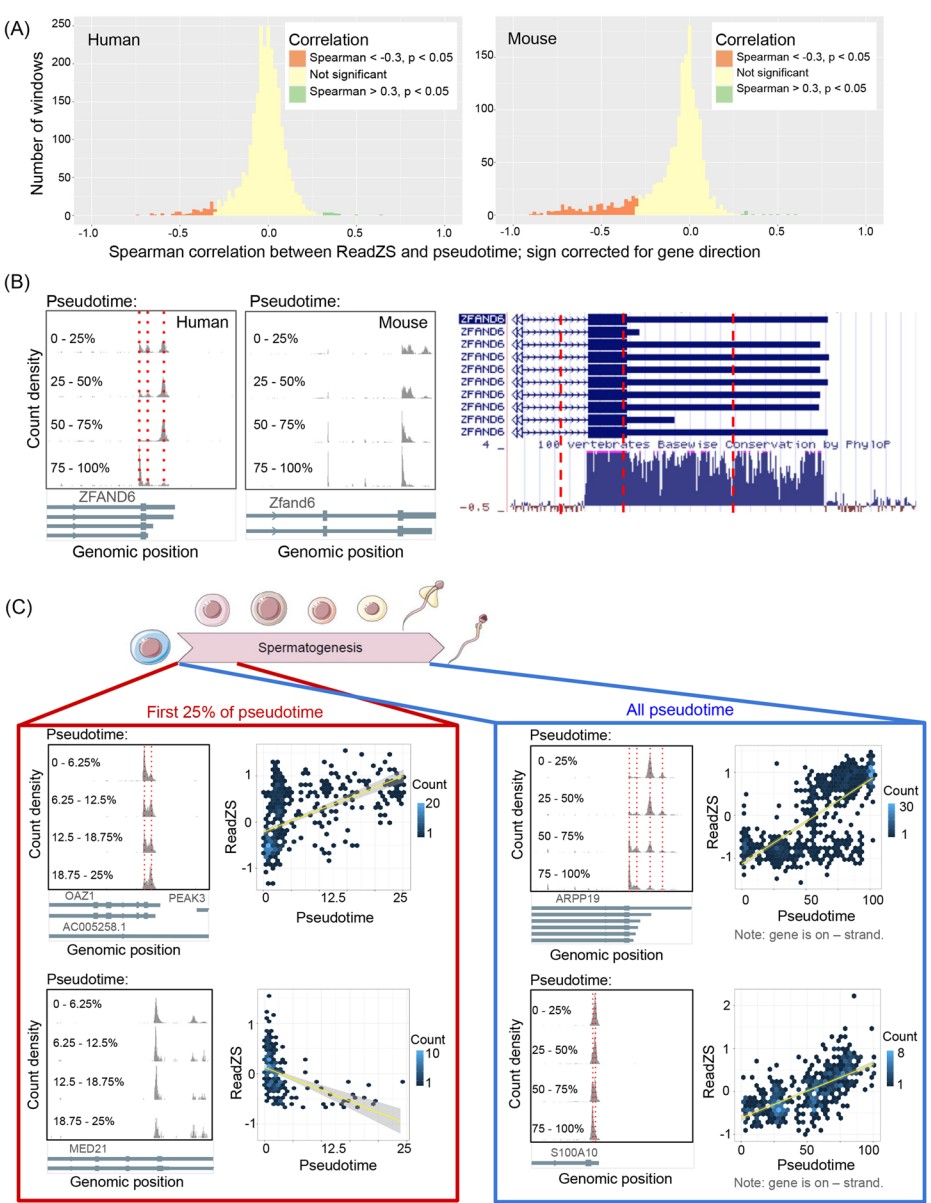

**Fig. 2** (See legend on previous page.)

and thus would likely be missed by a peak caller (Fig. 2C). Other examples of pseudotime-correlated RNAP include windows with overlapping or multiple peaks, e.g., windows covering the 3′ UTRs of *TSSK1B* and *SLC25A37* (Additional file 1: Fig. S5).

Manual curation of spermatogenic transitions can categorize sperm into developmental categories: spermatocytes, spermatids, and mature sperm. These stages have been pseudobulked to enable differential APA analysis [45]. However, because the ReadZS value is computed at a single-cell level, it potentiates discovery of fine-scale developmental transitions such as pseudo-temporal trends within immature sperm. To illustrate this capability, we used ReadZS to study differential RNAP within narrow windows of pseudotime. We correlated the ReadZS values to pseudotime, restricted to the earliest 25% of time in human (Fig. 2C). Out of 1433 windows with calculable

correlation in that pseudotime interval, 38 windows had significant correlation to pseudotime (|Spearman's correlation| > 0.3, Bonferroni-corrected *p*-value < 0.05). These windows include *OAZ1*, a gene implicated in ovarian function but with unreported regulation in sperm, which shows 3′UTR lengthening (Fig. 2C) [46]. Thirty-two windows had significant correlation within the first 25% of pseudotime but not over the entire range of pseudotime. These windows include *MED21*, a component of the mediator complex involved in transcriptional regulation which shows general shortening. Like *OAZ1*, *MED21* contains overlapping peaks and peaks at unannotated 3′ UTR sites, which may hinder a peak calling algorithm (Fig. 2C). Such discoveries highlight the power of a single-cell measure of RNAP that can discern RNAP within a "cell type," including events early in spermatogenesis.

### ReadZS discovers developmentally regulated RNAP in Arabidopsis root development

In mammals, 3′ UTR length has been shown to play a pivotal role in development, with cells producing longer transcripts over the course of embryonic development [10], and proliferating cells producing shorter UTRs that bypass miRNA-potentiated growth inhibition [9, 47]. APA in plants is also highly prevalent, with over 75% of transcripts undergoing APA [48], and highly regulated, including in the developmental process of flowering [49, 50]. Furthermore, the growth hormone auxin has been shown to affect APA, though the precise mechanisms remain unknown [51]. To determine whether APA is also regulated during root development, we applied the ReadZS to four 10X libraries of Arabidopsis root [52] (library names sc_1, sc_9_at, sc_10_at, and sc_11 from [53]). We calculated the correlation between ReadZS and pseudotime within each cell type. We then selected genomic windows that overlapped with annotated 3′UTR regions and counted how many significantly regulated windows (|Spearman's correlation| > 0.1, Bonferroni-corrected *p*-value < 0.05) had signs of correlation consistent with 3′UTR lengthening or shortening. Across the four libraries analyzed, 1047 window-cell type pairs had signs of correlation consistent with lengthening, while only 133 had signs consistent with shortening (Fig. 3A; Additional file 4: Table S3). To test if this result was affected by the incompleteness of 3′ UTR annotations, we recalculated the ReadZS for these libraries using gene positions instead of 1-kb windows, so that a ReadZS value was calculated for each cell and gene pair, instead of for each cell and genomic window pair as in the standard ReadZS workflow ("Methods"). After calculating correlation between ReadZS and pseudotime, we again found a general trend of lengthening, with 1763 gene-cell type pairs having correlation signs consistent with lengthening and only 302 consistent with shortening. These congruous results support the finding that the ReadZS detects transcript lengthening in Arabidopsis root development and differentiation, analogously to the 3′UTR lengthening observed in mammal development.

To identify individual genomic windows undergoing changes in RNA processing over the course of differentiation, we ranked window-cell type pairs by the magnitude of the Spearman's correlation between ReadZS and pseudotime, as done in the spermatogenesis analysis. In the most highly correlated window from one of the libraries, read distribution shifts occur over the 3′ end of gene *At5g10430* in atrichoblasts, suggesting that the changes in read distribution could be caused by APA in this gene (Fig. 3, Additional file 1: Fig. S6). This window is also significantly correlated with pseudotime in the cortex,

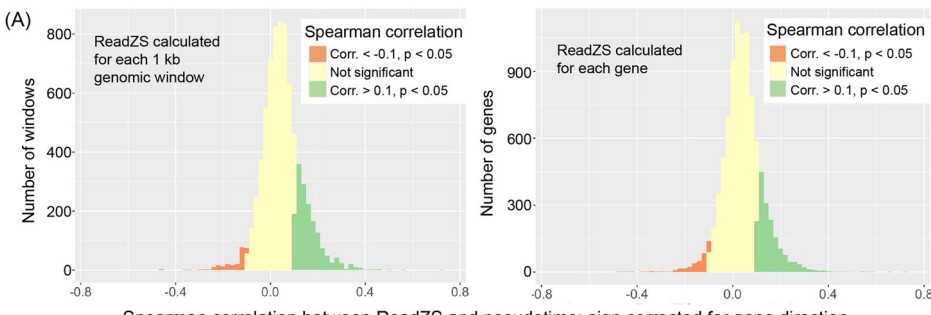

**Fig. 3** ReadZS detects regulated RNAP in Arabidopsis root cell differentiation. **A** The ReadZS detects a global trend of 3′ UTR lengthening in the pseudotime trajectory by cell type of Arabidopsis root cells, as indicated by significant positive correlations between ReadZS and pseudotime. The trend is consistent whether the ReadZS was calculated for each 1kb genomic window (left) or for each gene (right). This analysis was performed for each cell type in libraries sc_1, sc_9_at, sc_10_at, and sc_11, and all the correlation values are aggregated in the histogram. **B** Binned histogram of read positions from genomic windows in Arabidopsis root with significant correlation between ReadZS and pseudotime: in atrichoblasts, genomic window overlapping the gene *At5g10430* (Spearman's correlation = 0.658, Bonferroni-corrected *p*-value < 0.0001), from library sc_9_at. **C** Binned histogram of read positions from genomic windows in Arabidopsis root with significant correlation between ReadZS and pseudotime: in trichoblasts, genomic window overlapping the gene *At5g44020* (Spearman's correlation = 0.620 Bonferroni-corrected *p*-value < 0.0001), from library sc_11

trichoblast, procambium, and endodermis cell types in the same library. *At5g10430* is an arabinogalactan-protein known to be involved in reproduction, but the role of alternative UTRs is unknown. Furthermore, we observe reads downstream of the annotated 3′ end of the transcript, indicating that the existing annotation is incomplete.

The second most highly correlated window in library sc_11 occurs in trichoblasts and overlaps with the 3′UTR of gene *At5g44020*, for which only one isoform is known (Fig. 3, Additional file 1: Fig. S6). The differences in read distribution over the course of pseudotime are mostly in the second exon of the gene (31% of reads in the first 25% of pseudotime are from this exon, versus 7% of reads in the last 25% of pseudotime), suggesting developmentally regulated changes in splicing kinetics or intron retention. The ReadZS discovers both examples of regulated RNAP changes as it does not rely on peak calling or annotation.

### ReadZS has complementary power compared to other algorithms

To the best of our knowledge, no published method is comparable to ReadZS, which can predict novel APA sites and detect alternative RNAP using only 10X data, completely agnostic to annotation. We still view it as important to illustrate how ReadZS compares to other methods.

First, we compared ReadZS to Sierra [16], which uses pseudobulk analysis to detect differential transcript usage (DTU) including from fibroblasts in injured and uninjured mouse hearts [54, 55]. Sierra was used to measure 3′ UTR length changes between actively cycling fibroblasts (F-Cyc, F-Act, or F-CI) and resting fibroblasts (F-SL and F-SH) and found 631 genes exhibiting DTU (though with unknown FDR). We performed ReadZS analysis on this data and found 308 significant windows, across 272 genes (FDR < 0.05; Additional file 5: Table S4; "Methods"). Surprisingly, over 90% of these genes were not called or reported by Sierra. Restricting to 631 genes with DTU reported by Sierra, 126 had sufficient per-cell read coverage to calculate the single-cell-resolved ReadZS and 23 (18%) of these genes were called by ReadZS. Out of the 7 genes the authors investigated via RT-qPCR, only windows intersecting *CD47* and *COL1A2* had sufficient read depth ($\geq 5$ counts in $\geq 10$ cells) in at least two fibroblast populations to calculate median ReadZS values, and both were called significant by ReadZS. Despite the limitation of shallow read depth, ReadZS discovers many cases of regulated 3′ UTR changes missed by Sierra. Examples of new discoveries by ReadZS include *Rpl13a*, missed by Sierra despite having two cleanly separated APA peaks, and *Rab2a*, where multiple APA sites in the final exon create overlapping peaks which we expect to be missed by peak calling-based methods (Fig. 4A). This analysis illustrates that ReadZS is a complementary approach that recovers genes found by other algorithms and reveals biology they miss.

Next, we compared ReadZS to MAAPER, a model-based probabilistic approach for predicting polyadenylation sites in data and identifying APA [17]. The main limitation of MAAPER is that it requires an existing database of polyadenylation sites; therefore, it cannot detect novel APAs and cannot even be applied to emerging model organisms without any or at most partial annotations such as Arabidopsis. When applied to single-cell RNA-seq data, MAAPER uses pseudobulking to perform pairwise comparison between cell types, so we performed pairwise comparisons with ReadZS as well, even though this is not the normal use case of ReadZS. We note that the statistical framework and workflow of MAAPER are not designed for running on separate single cells, so we did not use SS2 data to compare ReadZS and MAAPER. To set up the pairwise comparisons, we selected five cell types shared between HLCA P2 and P3. Then, we ran each algorithm on every possible pair of cell types from P2 and P3. As there is no ground truth for regulated RNAP differences in HLCA, we defined three proxy measures to evaluate the performance of the algorithm: one proxy for true positive rate (TPR), and two proxies of false positive rate (FPR1 and FPR2; see "Methods"). We should note that our proxies for estimating true positive rate and false positive rate rely on the assumption that there are no true biological differences between the two HLCA individuals, which is of course impossible. As such, we expect to always detect some non-zero level of "false positive" as defined by our measures, reflecting actual differences between the individuals. In these comparisons, ReadZS

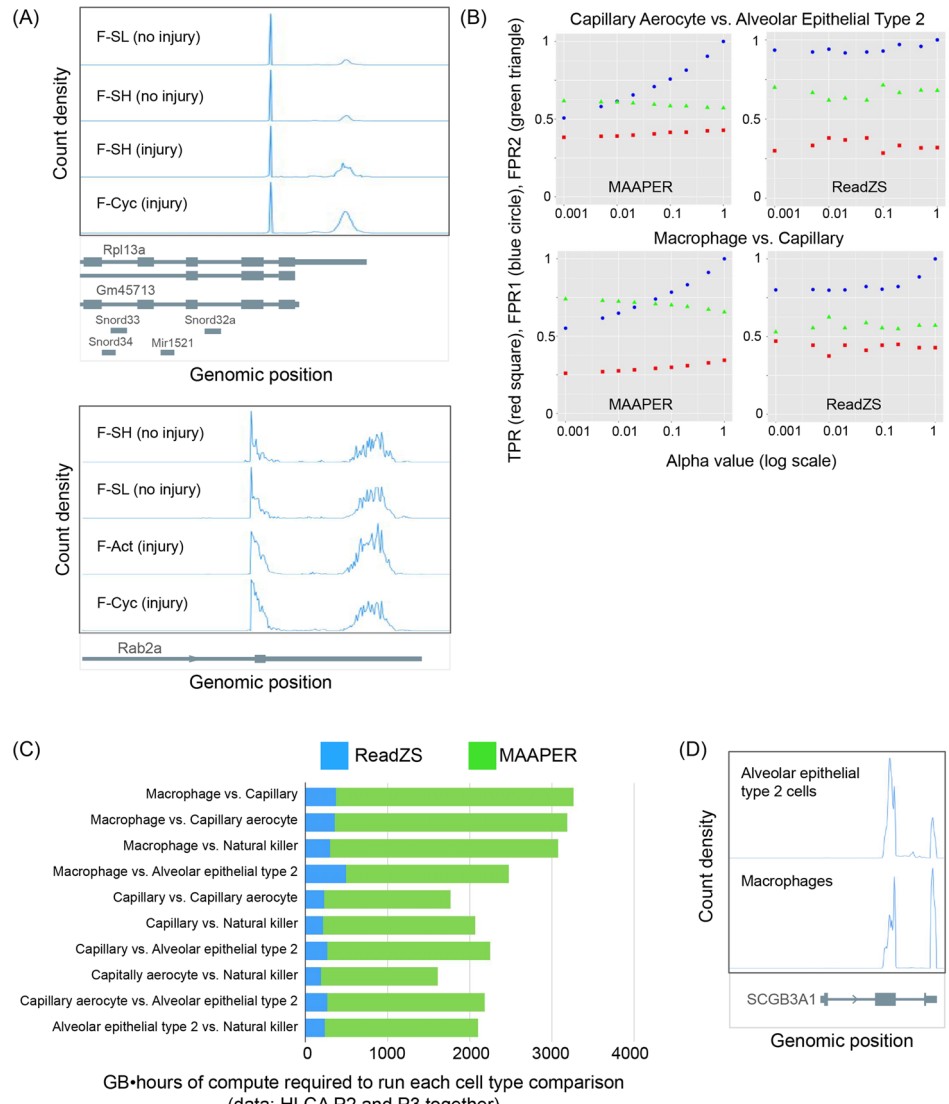

**Fig. 4** Comparison of ReadZS with pseudobulk-based approaches for APA detection. **A** The ReadZS has unique power to discover regulation missed by peak callers. In both of these examples called by ReadZS as significant, the change in relative peak height suggests the different groups of fibroblasts use alternate PA sites at different rates, yet these genes were not called by Sierra as undergoing DTU. Top: Rpl13a has cleanly separated peaks but was not called by Sierra. Bottom: the jagged peaks of reads in the 3'UTR of Rab2a might hinder peak calling-based methods. **B** Proxy measurements for true and false positive rates of MAAPER and ReadZS, from comparisons of different pairs of cell types from HLCA P2 and P3, evaluated at different alpha values (corrected *p*-value cutoffs for significance): "true positive rate" proxy (TPR)—red squares; "false positive rate" proxy 1 (FPR1)—blue circles, and "false positive rate" proxy 2 (FPR2)—green triangles (see "Methods" for calculation of these metrics). Missing points indicate that there were not sufficient significant genes or windows to calculate the proxy measurement. Full set of plots is in Additional file 1: Fig. S5. **C** Comparison of the total memory and run time (measured in GB•hours) for running ReadZS or running MAAPER on the same data. Each row represents a set of data, specifically a comparison between two cell types subsetted from the combined HLCA P2 and P3 datasets. ReadZS GB•hours were calculated automatically for each Nextflow run in the Nextflow Tower interface. MAAPER GB•hours were calculated by multiplying the memory allocated to each job with the number of hours required to run the data. **D** *SCGB3A1* is one of many genes not called by MAAPER but identified by ReadZS as undergoing cell type-specific RNAP. The window overlapping *SCGB3A1* was called as significant by ReadZS in the comparison between macrophages (from HLCA P2 and P3) and alveolar epithelial type 2 cells (from HLCA P2 and P3)

was able to achieve similar TPR, FPR1, and FPR2 as MAAPER, despite not relying on pseudobulking as done in MAAPER (Fig. 4B, Additional file 1: Fig. S7). Furthermore, examining the read distribution of several windows called by ReadZS as significant but considered "false positives" in this analysis, we observe differences in read distribution at 3′UTR sites, suggesting that the two individuals may have true differences in polyadenylation for certain genes. For example, a window overlapping the 3′UTR of the gene *TNFSF10* was called as significant by ReadZS when comparing P2 capillary cells and P3 capillary cells (Additional file 1: Fig. S8). Although this would count as a "false positive" by our FPR1 definition, the read distribution indicates a real difference in isoform usage between the two individuals (Additional file 1: Fig. S8), indicating that this is in fact computationally a true positive. Furthermore, this same genomic window was called as significantly differentially regulated between epithelial type II pneumocytes and capillary endothelial cells in lung data from Tabula Sapiens (unpublished data), confirming that *TNFSF10* undergoes regulated RNAP in lung.

Because the ReadZS score is calculated at the single-cell level, in a pairwise comparison ReadZS will detect fewer differential RNAP events compared to a pseudobulking-based such as MAAPER, which aggregates all the reads across individual cells within a cell type. Accordingly, ReadZS called fewer genes as having significant differences in RNAP: across all the comparisons run, 1358 calls (unique celltype1 - celltype2 - gene) were found by ReadZS only, 163,085 calls were found by MAAPER only, and 588 calls found by both (43% of the genes found by ReadZS), indicating that many of the ReadZS calls are potential true positives. ReadZS was also able to detect clear cases of APA missed by MAAPER, such as in the gene *SCGB3A1* (Fig. 4D). We note that MAAPER also required much more memory and time than ReadZS to run the same comparisons (Fig. 4C). Overall, this analysis demonstrates that ReadZS—despite not using pseudobulking—can achieve similar levels of sensitivity and specificity when compared to a state-of-the-art algorithm designed to detect APA.

Finally, we compared ReadZS to scDaPars, an algorithm for quantifying APA at single-cell resolution [15]. We used the same five cell types from the HLCA data sets as used in the MAAPER comparison, and similarly ran scDaPars on every possible pair of data set 1/cell type A vs. data set 2/cell type B. scDaPars requires several preprocessing steps, namely splitting 10X data into separate files for each cell, converting BAMs to wig files, and running DaPars2, before running scDaPars ("Methods"). Though we were able to run DaPars2 on each cell type pair, there were not sufficient APA events to run scDaPars successfully for any pair. We note that ReadZS is able to detect APA events at a true single-cell level and is thus not constrained by smaller data sets.

A pseudobulking-based method (such as Sierra and MAAPER) will tend to have greater power in pairwise comparisons than a true single-cell-resolved method such as ReadZS. However, ReadZS has power against alternatives where similar methods lack it: (1) what ReadZS can detect is not restricted to the 3′ UTR; (2) multiple APA sites within the same exon of a gene can create overlapping peaks in the read coverage, which cannot be quantified by a peak calling method (e.g., Fig. 4A); and (3) because ReadZS is a true single-cell measure of differential RNAP, it can automatically discover RNAP regulated as a function of pseudotime, which no other method is capable of. Importantly, discoveries by ReadZS include shifts in read distributions that would be missed by published

methods [15, 16, 45] for APA detection based on peak calling. Finally, ReadZS does not depend on any gene or polyadenylation annotation for detecting regions with significant RNAP, so ReadZS can be applied to genomes with incomplete or missing annotations.

## Discussion

In summary, ReadZS is a new statistical approach that does not involve pseudobulking, imputation, or peak calling. ReadZS does not use any annotations, such as gene boundaries or polyadenylation sites, to identify regions of the genome with regulated RNAP. As a true single-cell measure of differential RNAP, ReadZS can be integrated with continuous metadata such as pseudotime to identify developmentally regulated, continuous changes in RNAP—which cannot be done with any existing methods to detect RNAP. Unlike other methods, ReadZS can be directly applied to droplet-based scRNA-seq such as 10X Chromium (10X) without needing separate sequencing files for each single cell (as in scDaPars [15]) or for each cell type being compared (as in Sierra [16] and MAAPER [17]). To the best of our knowledge, there is no other computational method that is both pseudobulking- and annotation-free, does not rely on peak calling, and can provide true single-cell quantification for RNAP and provide a well-defined statistical criterion for identifying regulated cell type-specific RNAP events at a controlled false discovery rate (Table 1).

ReadZS quickly discovers novel regulated RNAP in a variety of contexts. Applying the ReadZS to 10X data from the HLCA, we discovered novel RNAP regulated at the cell type level, including the use of previously unannotated 3′ UTRs. ReadZS showed highly consistent results between the three individuals in the HLCA dataset. To demonstrate the utility of ReadZS in conjunction with continuous metadata, we applied ReadZS to paired datasets of human and mouse spermatogenesis. We observed global 3′ UTR shortening in mouse, which has been previously documented [37, 38], but we also found the first evidence of global 3′ UTR shortening in human spermatogenesis. Furthermore, by comparing the genes called as significant by ReadZS in the two different organisms, we found significant evolutionary conservation of genes undergoing regulated 3′ UTR changes. We also applied ReadZS to a highly studied but relatively less well annotated model organism, Arabidopsis. In a dataset of root development, we observed global 3′ UTR lengthening over the course of development. Global 3′ UTR lengthening is known to occur over mammalian development, with shorter 3′ UTRs in proliferating cells thought to evade miRNA-based inhibition [9, 47], but such a phenomenon is completely novel in plants.

The computational pipeline is efficient and lightweight and can be easily integrated into any existing single-cell pipeline. The window sizes used in this manuscript and the use of poly(A) primed data are not necessary for the methodology developed here. For example, windows could be chosen adaptively or based on a subset of features of interest. In addition, 5′ capture technology, SS2 data, or even scATAC-seq [56] could all be used as inputs to ReadZS because the algorithm operates on read distributions that need not form peaks. For example, we would expect the ReadZS could detect differential 5′ UTR use, intron retention, or exon inclusion when windows include these features along with neighboring features that are

not differentially processed (e.g., a constitutive exon or UTR). We anticipate that ReadZS should also be a powerful analytic tool for data such as that generated by single-nucleus sequencing or derived from split-pool tagging [57]. A final area of future work will be to test whether more cell types and states can be defined when the ReadZS value is used—by itself or in conjunction with gene expression—to perform clustering analysis or trajectory inference.

## Conclusion

ReadZS is a novel, reproducible, robust, and annotation-free statistical algorithm to detect regulated RNAP in high-throughput single-cell RNA sequencing data. Applying it to primary cells reveals new biology of RNAP, including in regions outside and within the 3′ UTR and encompassing regulation missed by peak calling algorithms. We anticipate that further analysis of the ReadZS will facilitate deeper functional inference for regulated RNAP in single cells, including 3′ UTR use. As more single-cell RNA-seq data becomes available for poorly annotated or non-model organisms, annotation-free approaches are increasingly critical for discovering regulated RNAP.

## Methods

### Creating counts tables from 10X BAM

The ReadZS summarizes the transcription state of a genomic window in a single cell. It is calculated using only reads that fully align to the genome with no gapping, so it excludes spliced reads. 10X reads were aligned using STAR (v 2.7.5.a) [58] with default parameters except for chimSegmentMin = 10 and chimJunctionOverhangMin = 10. UMI demultiplexing and cellular barcode identification and correction for 10X data was performed using UMI-tools [59]. BAM files were opened with Samtools and reads were filtered based on the CIGAR string "<length(SEQ)>M" and MAPQ score 255 to only allow uniquely mapping exact and full-length matches. The reads were then split by chromosome and strand. The reads were deduplicated, by removing cells with any duplicated UMIs or UMIs aligning to more than one unique position. The reads were then collapsed using the identifier column and counted at each position.

Each chromosome is split up into equal-sized windows with size inputted by the user—5-kb windows were used in human and mouse analysis and 1-kb windows were used in Arabidopsis analysis. Each read is assigned a stranded window based on the read's position and strand. The tables of reads and counts are then separated by chromosome. If there are multiple samples or files within the experiment, the counts tables from the same chromosome from different samples are concatenated together, e.g., all reads from chromosome 1 from any file are in the same counts file. For HLCA data, the data was divided by participant for ReadZS calculation in order to avoid any batch effects between individuals. For the Sierra, human spermatogenesis, and mouse spermatogenesis datasets, all data was analyzed together. For Arabidopsis data, each BAM file was analyzed separately.

**ReadZS calculation for each window and cell type**

To calculate the ReadZS value for a window in a particular cell $i$, the genomic positions falling within the window across all cells are assigned an increasing rank, with the most $5'$ and $3'$ positions (with at least one aligned read) assigned rank 1 and the highest rank, respectively. Only positions appearing in the data (i.e., positions with at least one read mapped there) are assigned a rank value and there is no gap in ranks between one position and the next, even if the positions are far apart on the genome.

For each given window, let $m_{ir}$ be the number of aligned genomic reads at the position with rank $r$ along the window in cell $i$. The weighted ranks in each cell are found by multiplying each rank $r$ by the number of reads at that rank, $m_{ir}$, in cell $i$. If the total number of reads within the window across all cells is $N$, we can compute $\mu$, the mean rank for this window across all cells, as:

$$\mu = \frac{\sum_i \sum_r r \; m_{ir}}{N}$$

and the standard deviation $\sigma$ of the weighted ranks by:

$$\sigma = \sqrt{\frac{1}{N}\sum_i \sum_r m_{ir}(r - \mu)^2}$$

The ranks within the window are renormalized by subtracting the mean $\mu$, and dividing by the standard deviation of the ranks, $\sigma$:

$$\tilde{r} = \frac{r - \mu}{\sigma}$$

Finally, the ReadZS value $z_i$ for the window in cell $i$ is computed as the weighted average of the normalized ranks:

$$z_i = \frac{\sum_r m_{ir} \tilde{r}}{N_i}$$

where $N_i$ is the total number of reads from the window in cell $i$. It can be seen that the expectation of $z_i$ is zero and its variance is approximately $\frac{1}{N_i}$, knowing that the variance of the sum of independent random variables equals the sum of their variances. This is an approximation as we assume that the ranks of the aligned reads across a window are independent.

**Identification of windows with regulated RNA processing: median ReadZS and its p-value for each window/cell type pair**

When cell type metadata is available, cells can be assigned a cell-specific annotation (e.g., lung macrophage). For each window, a median ReadZS is then calculated within each cell type. To calculate the median ReadZS for a pair of a window and cell type, we required a minimum of 20 cells with at least 10 counts in that window-cell type combination. In the HLCA data, lowering thresholds for calculating ReadZS to 5 counts in 10 cells results in 2578 windows (across 2160 genes) with calculable ReadZS and 374 windows (14.5%) called as significant likely due to decreased statistical power resulting from fewer reads and cells. In P1 and P2, respectively, at lower count thresholds, 112 (resp. 403) windows, 10.3%, (resp.,

15.1%) were significant out of 1084 (resp. 2672) calculable. For the mouse fibroblast data, we reduced these minimum requirements to 10 cells with at least 5 counts to account for the lower read depth. To systematically prioritize windows for further follow-up studies, windows were ranked according to the range of median ReadZS values across all cell types. To find which genes and 3′ UTRs intersect these windows, we intersected the window positions with annotation BED files of genes and 3′ UTRs requiring an overlap of at least 25% to annotate a window with that gene or 3′ UTR.

To evaluate whether there is a significant difference that is more extreme than expected by chance between the median ReadZS values for a window across different cell types, we compute a *p*-value by adopting an approach from [60] that was also used for the SpliZ method [61]. For each window being present in $I$ cell types, let $n_i$ be the number of cells in cell type $i$, $\theta_{n,i}$ be the median of the ReadZS values across cell type $i$, and $\sigma^2_{n,i}$ be the sample variance of the ReadZS values for the window from cell type $i$. We can compute the following test statistic on the ReadZS medians for the window:

$$T_{n,1} = \sum_{i=1}^{I} \frac{n_i}{\hat{\sigma}^2_{n,i}} \left[ \hat{\theta}_{n,i} - \frac{\sum_{i=1}^{I} n_i \hat{\theta}_{n,i} / \hat{\sigma}^2_{n,i}}{\sum_{i=1}^{I} n_i / \hat{\sigma}^2_{n,i}} \right]^2$$

To obtain a null distribution for this test statistic, we permute the cell type assignments a number of times (we have used 100 permutations) and then compute the test statistic $T^{(j)}_{n,1}$ for each permutation. According to [60], this permutation distribution converges to the chi-squared distribution with $k-1$ degrees of freedom. So, before starting with permutations, we first calculate the *p*-value for each window ($p_\chi$) by comparing it to the chi-squared distribution. This approach saves compute time by allowing us to quickly filter out most windows that are not significant. Then, only if the window has $p_\chi < 0.05$, a *p*-value based on permutations ($p_{perm}$) is computed by permuting cell type labels. To determine whether the observed test statistic $T_{n,1}$ is extreme in either direction, we first compute the cumulative distribution function (CDF) of the permutation distribution as:

$$\text{cdf}_{\text{perm}} = \frac{\sum_{j=1}^{J} \mathbb{I}\left\{ T^{(j)}_{n,1} < T_{n,1} \right\}}{J}$$

and using this CDF, we can calculate a two-sided *p*-value as

$$p_{\text{perm}} = 2 \min \left( \text{cdf}_{\text{perm}}, 1 - \text{cdf}_{\text{perm}} \right)$$

To correct for multiple hypothesis testing across windows, we apply Benjamini-Hochberg procedure to the $p_{\text{perm}}$ values and use a significance level of 0.05 to obtain the list of significant windows with cell type-specific RNA processing.

### Identification of windows with regulated RNA processing: correlation between ReadZS and pseudotime

In the human and mouse spermatogenesis datasets, as well as the Arabidopsis root dataset, we calculated the correlation between ReadZS and pseudotime for all windows with at least five reads from that window in at least 300 cells. We called a window as

significant if it had |Spearman's correlation| > 0.3 (for spermatogenesis) or |Spearman's correlation| > 0.1 (for Arabidopsis), and Bonferroni-corrected $p$-value < 0.05.

### 3′ UTR shortening in human and mouse spermatogenesis

To examine changes in the 3′ UTR length, we intersected all the significantly correlated genomic windows in the human and mouse spermatogenesis datasets with RefSeq 3′UTR annotations obtained from the UCSC Table Browser, requiring an overlap of at least 25% to annotate a window with that 3′ UTR. We then determined the "sign-corrected correlation value" by multiplying the Spearman's correlation coefficient by −1 if the genomic window was on the minus strand. That way, a window with a negative correlation to pseudotime always indicates a skew towards more upstream reads for that gene, i.e., 3′ UTR shortening if the window covers a 3′ UTR region. We first considered genomic windows with any 3′ UTR annotations: in human, we found 79 windows (85%) out of 93 with negative sign-corrected correlations, consistent with 3′ UTR shortening; in mouse, we found 307 (99%) out of 310 windows with negative sign-corrected correlations. Since a genomic window may contain several exons or even several genes, we also considered genomic windows with 3′ UTR annotations but without any 5′ UTR or exon annotations. Among those windows, we found 6 (86%) out of 7 in human and 66 (96%) out of 69 in mouse had negative sign-corrected correlations, consistent with overall 3′ UTR shortening.

### Overlap between human and mouse windows with significant correlation

To test whether there were genes exhibiting similar changes in RNA processing over spermatogenesis in both mouse and human, we selected the windows from both data sets with calculable correlation to pseudotime (requiring a minimum of 5 counts per window per cell in at least 300 cells) and intersected the two data sets to find windows with matching RefSeq gene names in mouse and human. Among window pairs with the same gene name in human and mouse (374 in total), we found 56 window pairs where both windows were significantly correlated with pseudotime, and 40 of those had negative correlation values for both windows (after correcting for gene direction). We used a hypergeometric test to calculate whether these overlaps of significance and correlation sign were more extreme than expected by chance, using the R function phyper.

### 3′ UTR lengthening in Arabidopsis root

We applied the ReadZS to libraries sc_1, sc_9_at, sc_10_at, and sc_11 from [53] and calculated correlation with pseudotime for each window and cell type with at least five reads from that window in 300 cells of that cell type. To examine changes in the 3′ UTR length, we intersected the genomic windows with 3′UTR annotations obtained from the Araport11 genome release GFF file from arabidopsis.org, requiring an overlap of at least 25% to annotate a window with that 3′ UTR. We then determined the "sign-corrected correlation value" by multiplying the Spearman's correlation coefficient by −1 if the genomic window was on the minus strand. That way, a window with a negative correlation to pseudotime always indicates a skew towards more upstream reads for that gene, i.e., 3′ UTR shortening if the window covers a 3′ UTR region. Out of 1180 window-cell

type pairs with |Spearman's correlation| > 0.1 and Bonferroni-corrected *p*-value < 0.05, there were 1047 (88.8%) windows with positive sign-corrected correlations, consistent with 3′UTR lengthening.

We also created a version of the ReadZS that uses gene regions instead of genomic windows as the base unit for ReadZS calculation. Specifically, instead of assigning reads to equally sized genomic windows and calculating a ReadZS value for each cell and window, we assigned reads to genes and calculated a ReadZS value for each cell and gene. We ran this version of the ReadZS on the same four Arabidopsis libraries and then calculated correlation with pseudotime in the same way. Then, without further filtering the list of genes, we determined the sign-corrected correlation values and counted how many gene-cell type pairs had positive or negative sign-corrected correlations. Out of 2065 gene-cell type pairs with Spearman's correlation| > 0.1 and Bonferroni-corrected *p*-value < 0.05, there were 1763 (85%) with positive sign-corrected correlations, consistent with 3′UTR lengthening.

### Concordance of ReadZS between pairs of data sets

In order to assess whether the windows called as significant by ReadZS have consistent cell type-specific regulation of RNA processing between data sets, we created a test statistic that measures concordance in ReadZS values for a window between data sets. Assume that for a genomic window called as significant, cell type "A" has a higher median ReadZS value compared to cell type "B," i.e., the read distribution in cell type A is more skewed upstream relative to cell type B. If these differences in read distributions between cell types reflect real biological signals, we expect cell type A to consistently have a higher median ReadZS than cell type B in different biological replicates. Therefore, for each genomic window called as significant, we expect the cell types to follow the same ranking as determined by their median ReadZS values. Accordingly, we created the following test statistic to measure the concordance between two data sets for each significant window:

$$x = \sum_{j=1}^{N} \frac{1}{mj} \sum_{i=1}^{mj} \left| R_{ij} - R_{ij}{}' \right|,$$

where $R_{ij}$ is the rank of the *i*th cell type out of $m_j$ cell types for genomic window *j*, $R_{ij}{}'$ is the rank of the same cell type and window but in the second data set, and *N* is the total number of significant genomic windows. If most windows have similar rankings of cell types in the two data sets, the differences in ranks between the data sets will tend to be small, resulting in a smaller value for *χ*. We simulated a null distribution for *χ* for each pair of data sets by calculating *χ* 5000 times using permuted ranks. For each iteration, we first randomly permuted the ranks of cell types in the second data set, and then we used the intact first data set and the permuted second data set to compute *χ*. For each pairwise comparison of data sets, we were then able to calculate a *p*-value by comparing the real value of *χ* against the simulated null distribution.

### Comparison with MAAPER

We ran MAAPER on every possible cell type pair (P2 or P3 cell type A compared against P2 or P3 cell type B) from the below cell types in HLCA data:

- Lung (immune) macrophage
- Lung (endothelial) capillary
- Lung (endothelial) capillary aerocyte
- Lung (immune) natural killer
- Lung (epithelial) alveolar epithelial type 2

We defined the following measures to evaluate the performance of each algorithm:

(A) "True positive rate" proxy (TPR) was defined as the proportion, out of all windows (for ReadZS) or genes (for MAAPER) tested, of windows/genes that were found to undergo significant APA in both P2 cell type A vs. P3 cell type B, and in P2 cell type B vs. P3 cell type A, with the same effect direction in both comparisons. These windows/genes demonstrated differences between cell types from different biological samples, and these differences are replicated when the replicates are switched, suggesting that there are consistent cell type-specific differences in APA.

(B) "False positive rate" proxy 1 (FPR1) was defined as the proportion, out of all windows/genes tested, of windows/genes that were found to undergo significant APA in either the P2 cell type A vs. P3 cell type A comparison or the P2 cell type B vs. P3 cell type B comparison. These windows/genes were called as significant on the basis of differences detected between biological replicates, within the same cell type, suggesting that there is no cell type-specific APA occurring in these cases, so these are counted as false positives.

(C) "False positive rate" proxy 2 (FPR2) was defined as the proportion, out of all windows/genes tested, of windows/genes that were found to undergo significant APA in both P2 cell type A vs. P3 cell type B, and in P2 cell type B vs. P3 cell type A, but with a *different* direction of effect in the two runs. These windows/genes were called as significant on the basis of differences detected between cell types from different biological samples, that are not replicated when the replicates are switched, suggesting that these effects could be false positives.

**Comparison with scDaPars**

We used the same (as in the MAAPER comparison) set of all possible cell type pairs (P2 or P3 cell type A compared against P2 or P3 cell type B) from the below cell types in HLCA data:

- Lung (immune) macrophage
- Lung (endothelial) capillary
- Lung (endothelial) capillary aerocyte
- Lung (immune) natural killer
- Lung (epithelial) alveolar epithelial type 2

For each set of data, we did the following preprocessing steps in preparation for running scDaPars:

(1) Split the BAM files by cell ID, using awk

(2)  Converted the BAM files to wiggle files, using samtools

(3)  Created a table of wig file paths and corresponding numbers of mapped reads, for input into DaPars2, using samtools

(4)  Ran DaPars2 on the data, with each cell in a separate file (as indicated in the scDaPars GitHub page)

(5)  For each run, combined the separate output files for each chromosome into a single file

(6)  Ran scDaPars on each combined output file, which failed in every case due to insufficient points

### Peak calling using Gaussian mixture model

For each significant window called by ReadZS, we performed peak calling by fitting a Gaussian mixture model (GMM) to the distribution of the reads from the entire dataset across that window. We obtain the optimal number of components in the GMM, which corresponds to the number of peaks in the read distribution, as the knee point in the integrated complete-data likelihood (ICL) curve across different numbers of components. We apply the ICL criterion to the read distribution of each window that was called as significant, and the peaks are found via fitting a Gaussian mixture model. We further compute the Bhattacharya distance between the components. If the distance is <0.5, we reduce the number of peaks by one and again fit a GMM. We stop if there is only one component remaining or the distance between components is at least 0.5.

### Pipeline implementation using Nextflow

To allow for reproducible and parallelizable results, the ReadZS pipeline is written in Nextflow [25]. Nextflow is an open-source workflow management system that integrates command-line and scripting tools to analyze large-scale datasets. The ReadZS workflow takes in BAM alignment files from 10X or SS2, and it performs processing and calculation steps on all of the files in parallel. The workflow then outputs tables with cell type medians and their associated *p*-values (Additional file 1: Fig. S9). The workflow also allows users to input dataset-specific parameters, such as cell annotation files, genome window files, and the columns used to define ontology (cell type or other grouping). To further enhance portability, the entire workflow can be run on a high-performance computing platform or on a cloud computing platform.

### Calculating distance to 3′ UTR annotations

For human samples, the Gencode GFF3 files were used for distance calculations and plotting. For mouse samples, RefSeq GFF3 files were used. To extract 3′UTR regions in bed format, the GFF3 file was filtered for `feature type = 'three_prime_UTR'`, with the `ID` field used as the 3′UTR identifier. To extract gene regions in bed format, the GFF3 file was filtered for `feature type = 'gene'`, with the `gene_name` field used as the gene identifier.

To determine the `num_3UTR_300bp_downstream` column, a bed file was created for each window's start and end positions, shifted 300bp downstream relative to the strand of the window. To determine 3′ UTR ends, the 3′ UTR bed file was filtered for regions on

the same strand as the window, and the start position or end position was used to create a separate bed file for minus and plus strands, respectively. The command `bedtools intersect -c -s` was used to find the number of overlapping 3′ UTR ends in each of the shifted windows.

To determine the `window_has_gene` column, a bed file was created for each window. The command `bedtools intersect -c` was used to determine if there were any annotated genes intersecting the window. To determine the `peak_has_600bp_downstream_gene` column, a bed file was created for the region of each peak and 600bp downstream from the peak, relative to the strand of the window. If the peak was at position less than 600 and the window strand was "minus," then the bed file was created for the region from 0 to the peak. To determine if each shifted window intersects with any annotated genes, the command `bedtools intersect -c -s` was used for the shifted window and the annotated genes bed file.

To determine the closest upstream and downstream 3′UTR ends, a bed file was created for each peak, and `bedtools closest` was used to determine the 3′ UTR ends that were the least distant from each peak. For peaks located in a "plus" stranded window, the closest upstream 3′ UTR end and its distance were determined from the output of `bedtools closest -c` of the peak bed file and the strand-respective 3′ UTR ends bed file, with the `ignore downstream` flag to only capture upstream regions. The closest downstream 3′ UTR end was determined with the same command, but with the `ignore upstream` flag to only capture downstream regions. For peaks located in a "minus" stranded window, the same commands were used, but with the `ignore upstream` flag used for upstream regions and the `ignore downstream` flag used for downstream regions, in order to account for reverse strandedness.

### Plot generation for cell type annotated data

To investigate windows that showed a large range in median ReadZS values, histograms were plotted showing the number of counts at each genomic position. To plot each window, every ontology (i.e., cell type or other grouping) for that window was sorted by its median ReadZS value. The top 2 and bottom 2 ontologies were then chosen to be plotted for each window. For each ontology, pass-filter reads were extracted if they came from that window and from the cell barcodes associated with that ontology. These reads were then deduplicated, with positions rounded to a bin size of 10. Each position was then counted, by summing the number of reads at each position per ontology in that window. The count was then normalized by the total number of counts per ontology in that window, to produce a percent score.

To plot this percent score, the counts were read into Gviz [62], and positions without count values were imputed with 0. Each ontology was used to create a data track, with an *x*-axis range of the window start and window end. A respective genome GFF file was used to plot the gene region track, with the GFF "transcript" feature excluded, for visual clarity.

### CDF and histogram generation for peak distances to 3′UTRs

The GMM-annotated peak tables were used to create the overlaid CDF and histogram of peak distances closest to a downstream 3′UTR end. For each dataset, the table was filtered for `peak_has_600bp_downstream_gene == True` and `df.

downstream_3UTR_dist < VALUE`, where VALUE is some bound on the *x*-axis. Unless otherwise stated, all plots are made with `bins = 100` and VALUE=[200, 800000]. The histograms and CDF plots were made with the matplotlib `hist()` function with `density=True`. The CDF plots were also plotted with `cumulative=True` and `histtype='step'`. The quantiles were calculated with the pandas `quantile()` function, to determine the 25th, 50th, and 75th quantiles. The quantile cutoffs are visualized by the red-dotted lines.

## Supplementary Information

---

Additional file 1: Supplementary figures. This file contains all supplementary figures, as well as in-depth descriptions of the supplementary tables (Additional files 2, 3, 4 and 5).

Additional file 2: Table S1. This table contains the cell type-specific RNA processing events detected in the HLCA dataset.

Additional file 3: Table S2. This table contains the cell type-specific RNA processing events detected in the mouse fibroblast dataset.

Additional file 4: Table S3. This table contains the cases of regulated RNA processing detected in human and mouse spermatogenesis.

Additional file 5: Table S4. This table contains the cases of regulated RNA processing detected in Arabidopsis root development.

Additional file 6. Review history.

---

### Acknowledgements
We thank the Salzman Lab for useful discussion, especially Julia Oliveri and Robert Bierman for comments that enhanced the clarity of the document. We thank Sarthak Satpathy for early prototypes of processed BAM files that were used in early versions of ReadZS. We also thank Dr. Benjamin Cole for his assistance in obtaining the Arabidopsis 10X data and associated metadata.

### Review history
The review history is available as Additional file 6.

### Peer review information

### Authors' contributions
EM contributed to development of the ReadZS statistic, wrote code for the ReadZS and significance calculations, analyzed data using the pipeline, and contributed to writing the manuscript. KC wrote code for the ReadZS and significance calculations, implemented the algorithm as a Nextflow pipeline, and analyzed data using the pipeline. RD contributed to development of the ReadZS statistic, wrote code for the peak calling steps, and contributed to writing the manuscript. JS designed the ReadZS statistic, oversaw the project, and was a major contributor in writing the manuscript. All authors read and approved the final manuscript.

### Funding
E.M. is supported by the National Science Foundation Graduate Research Fellowship under Grant No. DGE-1656518 and a Stanford Graduate Fellowship. J.S. is supported by the National Institute of General Medical Sciences Grants R01 GM116847 and R35 GM139517 and NSF Faculty Early Career Development Program Award MCB1552196.

### Availability of data and materials
The ReadZS Nextflow pipeline along with detailed instructions and test data are available through a GitHub repository, under an MIT license [63]. The version of the code used to generate these findings is available through Zenodo [64]. The human lung scRNA-seq data used here was generated through the Human Lung Cell Atlas project [27] and is accessible through European Genome-phenome Archive (accession number: EGAS00001004344) [28]. Human and mouse unselected spermatogenesis data [39] was downloaded from the SRA database with accession IDs SRR6459190 (AdultHuman_17-3), SRR6459191 (AdultHuman_17-4), and SRR6459192 (AdultHuman_17-5) for human, and accession IDs SRR6459155 (AdultMouse-Rep1), SRR6459156 (AdultMouse-Rep2), and SRR6459157 (AdultMouse-Rep3) for mouse [40]. Arabidopsis root data [52] was downloaded from the SRA database with accession numbers SRR12046049 and SRR12046050 for library sc_1, SRR12046051 and SRR12046052 for library sc_9_at, SRR12046053 and SRR12046054 for library sc_10_at, and SRR12046055 and SRR12046056 for library sc_11 [53]. Mouse fibroblast data [54] was downloaded from ArrayExpress under identifier E-MTAB-7376 [55]. RefSeq annotations (used to annotate significant windows with intersecting genes and 3′ UTRs) were downloaded from UCSC Table Browser at genome.ucsc.edu; for GENCODE annotations (used when computing distances between peaks and 3′ UTRs), we used v37 for human and vM26 for mouse, downloaded from gencodegenes.org.

## Declarations

### Ethics approval and consent to participate
Not applicable.

### Competing interests
The authors declare that they have no competing interests.

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

## 