## [Additional file 6. Review history. · Genome Biology]

Review History

First round of review

Reviewer 1

Were you able to assess all statistics in the manuscript, including the appropriateness of statistical tests used? No

Were you able to directly test the methods? No

Comments to author:

In this manuscript, the authors designed an annotation-free and nonparametric strategy for single-cell measure of differential RNAP. This approach can provide a well-defined statistical criterion for identifying regulated cell type-specific RNAP events. However, the interpretation and explanation of some problems are still not clear enough for publication. Then I suggest it to be published in Genome Biology after major revision.

1. As mentioned in the second and fourth paragraph of the research background, the disadvantages of annotation-dependent and peak-calling approaches were described. What superiority do these approaches have? Are there any annotation-free approaches that can measure bulk-cell or single-cell RNAP?

2. As the authors mentioned, many annotation-dependent approaches have been developed in recent years. What does "annotation" refers to in theory and in application? How ReadZS achieves annotation-free?

3. The writing logic of the Background section seems not so reasonable. The third paragraph introducing the approach of this manuscript was inserted between two paragraphs of other approaches repeatedly.

4. Some writing and formal errors should be noticed. The paragraph format of Background section is inconsistent. And the position of Figure2A is not in place. Please re-check the format of the whole manuscript carefully.

Reviewer 2

Were you able to assess all statistics in the manuscript, including the appropriateness of statistical tests used? Yes: I don't see a problem with the statistical test.

Were you able to directly test the methods? No

Comments to author:

The authors propose a de novo finder for alternative polyadenylation (APA) events at the single-cell level from scRNA-seq data in this paper. The authors compare its performance with existing methods (MAAPER, Sierra, and scDapars). Furthermore, the proposed method is applied to actual data from human, mouse, and Arabidopsis.

The authors present the proposed method's principles, performance, and usefulness in clear text and diagrams. Although the proposed method has the advantage of finding novel APA events in the 3' UTR without gene annotation, it has the problem that the false positive rate is too high compared to other methods.

#1

The results in Figure 4 show that ReadZS has a very high FPR compared to MAAPER. Also, ReadZS's FPR does not decrease with decreasing alpha value, while MAAPER's FPR does. The authors should carefully consider these two points. For example, could you consider whether ReadZS's statistical methods or data analysis workflow may be vulnerable to data noise?

The scRNA-seq method using the template-switching method, including 10X Genomics Chromium, has the serious problem that reads are unfairly obtained from regions other than UTRs. Probably due to random mis-annealing of template switching oligo, intermediate parts of RNA, etc. are sequenced[1,2,3]. ReadZS does not use gene annotation and uses binned chromosomes without filtering, so it is I am concerned that ReadZS may detect such noise.

[1] <https://doi.org/10.1074/jbc.RA119.010676>

[2] <https://www.nature.com/articles/s41586-018-0414-6/figures/5>

[3] Extended Data Fig. 1 in [2]

I think this study will be an important insight into your quest to lower FRP. Also, please quantify the extent to which the proposed statistical method is affected by such noise. If such noise is a problem, why don't you incorporate a filtering step into ReadZS's workflow, for example, by using gene annotation?

The authors will also need to sort out the relationship between APA events that are false-positive and read distribution, genomic position, and genomic sequence features. The authors should use these features to determine the cause of the high FPR of ReadZS.

#2

How many were APA events detected in ReadZS and MAAPER, respectively? Of those, how many novel APA events were found in ReadZS?

#3

Which performs better with non-pseudo-bulking data, MAAPER or ReadRS?

4

Why are there large variations in ReadRS performance scores in Fig. 4 and Fig. S7? Is it because the predictions are used single-cell data? What data characteristics (e.g. library size) are associated with the score variation?

#5

Are methods other than ReadZS and MAAPER incapable of computing TPR and FPR?

We now address the comments made by the reviewers. Our responses to the reviewers' comments are in bold and the major changes in the revised manuscript are highlighted. We believe that the added analysis and new modifications have improved the quality of the manuscript and have addressed all of the reviewers' comments.

Response to Reviewer 1 comments:

In this manuscript, the authors designed an annotation-free and nonparametric strategy for single-cell measure of differential RNAP. This approach can provide a well-defined statistical criterion for identifying regulated cell type-specific RNAP events. However, the interpretation and explanation of some problems are still not clear enough for publication. Then I suggest it to be published in Genome Biology after major revision.

We thank Reviewer 1 for their helpful questions and suggestions and are grateful that Reviewer 1 has found this work to provide a well-defined statistical criterion for identifying cell-type-specific RNAP.

1. As mentioned in the second and fourth paragraph of the research background, the disadvantages of annotation-dependent and peak-calling approaches were described. What superiority do these approaches have? Are there any annotation-free approaches that can measure bulk-cell or single-cell RNAP?

We appreciate your comment. We have now clarified in the Background section (page 2, 1st paragraph) that pseudobulking increases power by combining single-cell data into bulk data, therefore pseudobulking can potentially discover more regulation when applied to pairwise comparisons of grouped cells (e.g. cell types). We note that - to our knowledge - there is no annotation-free method to detect APA from either bulk or single-cell data. All the existing methods for single-cell RNAP are annotation dependent, e.g. Sierra, MAAPER, and scDaPars all rely on gene annotations and would not be usable in an organism with poor or missing annotations. For example, mouse lemur (<https://doi.org/10.1101/2021.12.12.469460>) is an emerging model organism that has recently drawn enormous attention from genomics and evolutionary biology communities, but it has a poor genome annotation and no curated database of polyadenylation sites, so RNAP regulation in this organism could not be analyzed using existing methods.

2. As the authors mentioned, many annotation-dependent approaches have been developed in recent years. What does "annotation" refers to in theory and in application? How ReadZS achieves annotation-free?

Thank you for pointing this out. Our method does not require any annotations, such as gene boundaries based on genome annotation or a database of known polyadenylation sites, to identify regions of the genome with regulated RNAP. Instead, the ReadZS partitions the genome into equally-sized genomic windows, and tests for significantly regulated RNA processing differences within each window. We have now further clarified this in the revised text (Background section: page 2, paragraph 4; Results section: page 3, paragraph 6).

3. The writing logic of the Background section seems not so reasonable. The third paragraph introducing the approach of this manuscript was inserted between two paragraphs of other approaches repeatedly.

We appreciate your comment on the Background section. We have reorganized and revised the Background section to provide more context and better introduce our method (pages 1-2).

4. Some writing and formal errors should be noticed. The paragraph format of Background section is inconsistent. And the position of Figure 2A is not in place. Please re-check the format of the whole manuscript carefully.

We appreciate your comment. We have now fixed the formatting errors identified in the Background section and re-checked the format of the whole manuscript.

Response to Reviewer 2 comments:

The authors propose a de novo finder for alternative polyadenylation (APA) events at the single-cell level from scRNA-seq data in this paper. The authors compare its performance with existing methods (MAAPER, Sierra, and scDapars). Furthermore, the proposed method is applied to actual data from human, mouse, and Arabidopsis. The authors present the proposed method's principles, performance, and usefulness in clear text and diagrams. Although the proposed method has the advantage of finding novel APA events in the 3' UTR without gene annotation, it has the problem that the false positive rate is too high compared to other methods.

We thank Reviewer 2 for their helpful questions and suggestions and appreciate the reviewer's positive comments on the advantage of our method for finding novel APA events. We believe that by addressing the comments, our manuscript has improved significantly and the main strength of ReadZS which is being an annotation-free, peak-calling-free method with true single-cell-resolved measurements for APA is now better illustrated in the revised manuscript.

1. The results in Figure 4 show that ReadZS has a very high FPR compared to MAAPER. Also, ReadZS's FPR does not decrease with decreasing alpha value, while MAAPER's FPR does. The authors should carefully consider these two points. *[question and responses continued below]*

We appreciate your comment on the performance comparison between ReadZS and MAAPER. We have now clarified in the text (first paragraph of page 9) that as our comparison is based on the HLCA dataset, which is a real scRNA-seq dataset with no ground truth known. As such, the "TPR" and two "FPR" measures we used for comparison are not real true- and false-positive rates on "ground truth", but rather are proxy measurements (with calculation details fully explained in the Methods section: page 16, paragraph 2). Our proxies for estimating true positive rate and false positive rate rely on the assumption that there are no true biological differences between the two individuals in the HLCA dataset, which is of course impossible. As such, we expect to always detect some non-zero level of "false positive" as defined by our measures, reflecting actual differences between the individuals. Indeed, neither MAAPER nor ReadZS is able to achieve proxy "FPR" rates below ~0.25. The goal of this comparison is to assess how well ReadZS performs in the pairwise comparison scenario that a pseudobulking-based method is optimally designed for.

Furthermore, we investigated ReadZS calls that would count as "false positives" according to our proxy measurement: based on our FPR1 definition, these are genomic windows called as significant by ReadZS when comparing the same cell type between the two individuals. By visualizing the read buildup, we concluded that many "false positive" windows exhibited clear hallmarks of alternative polyadenylation, with read buildup forming peaks at annotated transcript 3' ends (highlighted in Supp. Figure 8). This suggests that while many of the calls between the same cell type across individuals might be considered "false positives" on the basis of our comparison method, they are in fact computational true positives.

1. *[cont]* For example, could you consider whether ReadZS's statistical methods or data analysis workflow may be vulnerable to data noise? The scRNA-seq method using the template-switching method, including 10X Genomics Chromium, has the serious problem that reads are unfairly obtained from regions other than UTRs. [...] Also, please quantify the extent to which the proposed statistical method is affected by such noise.
[question and responses continued below]

We have clarified in the text (Results: page 2, last paragraph, continued on page 3) that although this study is focused on 10X data and 3'UTRs/APA, ReadZS is not limited to APA analysis and can in fact detect changes in RNA processing in other parts of a transcript. Accordingly, reads originating from internal priming in 10X data do not contribute to noise; on the contrary, they are useful data that can be used to detect changes in RNA processing within the gene. ReadZS users interested in only reads originating from annotated 3'UTRs could restrict to only those genomic windows that overlap known 3'UTR regions.

1. *[cont.]* If such noise is a problem, why don't you incorporate a filtering step into ReadZS's workflow, for example, by using gene annotation? The authors will also need to sort out the relationship between APA events that are false-positive and read distribution, genomic position, and genomic sequence features. The authors should use these features to determine the cause of the high FPR of ReadZS.

We appreciate your comment and in fact agree that removing sequencing noise and reads originating from biochemical artifacts is a crucial step for avoiding false positives. As mentioned in the Methods section (last paragraph of page 11), the reads input to ReadZS are strictly filtered to only allow uniquely-mapping, exact, full-length matches to the genome. As such, we do not expect a high rate of sequencing error in the reads processed by ReadZS.

2. How many were APA events detected in ReadZS and MAAPER, respectively? Of those, how many novel APA events were found in ReadZS?

Because the ReadZS score is calculated at the single-cell level, in a pairwise comparison scenario ReadZS will detect fewer differential RNAP events compared to pseudobulk, which aggregates all the reads across individual cells within a celltype. Indeed, ReadZS called fewer genes as having significant differences in RNAP: summing across all the pairwise comparisons run, 1,358 calls (unique cell type 1 / cell type 2 / gene calls) were made by ReadZS and 163,085 calls were made by MAAPER. 588 (43%) of the calls made by ReadZS were also called by MAAPER, indicating that many of the ReadZS calls are potentially true positives. ReadZS was also able to detect clear cases of APA missed by MAAPER, such as in the gene SCGB3A1 highlighted in Figure 4D. We have reported these findings in the text (Results section: page 9, paragraphs 2-3).

3. Which performs better with non-pseudo-bulking data, MAAPER or ReadRS?

MAAPER relies on pseudobulking and cannot be run in a non-pseudobulking way (as shown in Table 1). In fact, the main strength of ReadZS over all existing methods is that it can provide true single-cell-resolved quantification for RNAP events. This single-cell resolution enables integration of RNAP quantification with other single-cell measurements, as highlighted by our study of developmentally-regulated RNAP in spermatogenesis - which can only be done through a method such as ReadZS with single-cell-resolved measures for RNAP.

4. Why are there large variations in ReadRS performance scores in Fig. 4 and Fig. S7? Is it because the predictions are used single-cell data? What data characteristics (e.g. library size) are associated with the score variation?

We appreciate the reviewer's concern about the error bars. These "error bars" in our previous submission were based on 95% binomial confidence intervals, which are directly related to the number of hits used in the proxy "FPR" and "TPR" calculations, rather than showing the performance variation at different iterations. Since the ReadZS called fewer genes as significant, the bars were larger for ReadZS compared to MAAPER. The bars did not actually represent any variation in our proxy measurements. Because these bars were misleading and gave the false impression of repeated measurements and high variability, we have now removed the bars in this updated version of the manuscript (Figure 4B and Supplementary Figure 7).

5. Are methods other than ReadZS and MAAPER incapable of computing TPR and FPR?

Thank you for your comment. We also tried to run the same comparison against scDaPars, but there were not sufficient APA events to run scDaPars successfully for any pair, and so we did not reach the step of calculating our proxy "TPR" and "FPR" measurements.

In general, there is no gold standard for computing TPR and FPR when comparing methods for APA, especially in single-cell data. Running methods on simulated data would allow us to compare to a "ground truth", but as far as we know, there is no available software to create simulated single-cell RNA-seq data that also has alternatively polyadenylated transcripts. When using real data to compare methods, there is no way to know the "ground truth," and different methods will identify different genes/transcripts. In a 2021 benchmarking study of APA methods for bulk RNA-seq data (<https://doi.org/10.1186/s13059-021-02502-z>), the authors used full-length long-read RNA-seq and 3' end-enriched RNA seq data as a "ground truth." However, long-read sequencing still has a much higher error rate than short-read sequencing, and single-cell long-read sequencing is very new and not widely available yet, making this type of benchmarking approach virtually impossible for comparing methods applied to single-cell data.

Second round of review

Reviewer 1

In the revised version, the authors addressed well most of the issues raised by the reviewers, and I am satisfied with these changes. Therefore, I suggest the acceptance of this manuscript.

Reviewer 2

I find that the authors answered most of the questions clearly. I think that the revised manuscript is acceptable for this journal.

However, the authors responded to Question 3 that MAPPER cannot be run on a single-cell level. In the MAPPER paper, They use a pseudo-bulk bam file to perform the calculations. However, could it be possible to run it by entering a single-cell bam file? Of course, the MAPPER paper does not show results for single-cell derived bam files. Also, the MAPPER model may not be limited to a probabilistic model for only bulk RNA-seq data. I am sure that is not what MAPPER's authors intended.

I am concerned that simply using MAPPER for a single-cell derived bam file may not perform similarly to ReadRZ. Even if MAPPER could achieve similar performance to ReadRZ, ReadRZ's features (annotation-free and peak-calling-free) would undoubtedly make it more valuable.

Therefore, I know that asking the ReadZS's authors to answer this question may be a bit excessive.

Authors Response

Point-by-point responses to the reviewers' comments:

Reviewer 2

I find that the authors answered most of the questions clearly. I think that the revised manuscript is acceptable for this journal.

However, the authors responded to Question 3 that MAPPER cannot be run on a single-cell level. In the MAPPER paper, They use a pseudo-bulk bam file to perform the calculations. However, could it be possible to run it by entering a single-cell bam file? Of course, the MAPPER paper does not show results for single-cell derived bam files. Also, the MAPPER model may not be limited to a probabilistic model for only bulk RNA-seq data. I am sure that is not what MAPPER's authors intended.

I am concerned that simply using MAPPER for a single-cell derived bam file may not perform similarly to ReadRZ. Even if MAPPER could achieve similar performance to ReadRZ, ReadRZ's features (annotation-free and peak-calling-free) would undoubtedly make it more valuable.

Therefore, I know that asking the ReadZS's authors to answer this question may be a bit excessive.

Response: In the original MAAPER publication, the authors only apply MAAPER to single-cell data by first pseudobulking by cell type. The authors do not apply MAAPER to individual single-cell BAMs, and MAAPER does not have any special mechanisms to deal with the scarcity of data obtained from individual single cells. As such, we do not think that MAAPER is designed - either in its probabilistic framework or in its implementation - to process single-cell BAMs such as Smart-Seq2 data. Using MAAPER in this way would constitute a different method which has not been tested or validated. We added a sentence to the manuscript to clarify this point.